# DLP 3D Printing Meets Lignocellulosic Biopolymers: Carboxymethyl Cellulose Inks for 3D Biocompatible Hydrogels

**DOI:** 10.3390/polym12081655

**Published:** 2020-07-25

**Authors:** Giuseppe Melilli, Irene Carmagnola, Chiara Tonda-Turo, Fabrizio Pirri, Gianluca Ciardelli, Marco Sangermano, Minna Hakkarainen, Annalisa Chiappone

**Affiliations:** 1Department of Fibre and Polymer Technology, KTH Royal Institute of Technology, Teknikringen 56-58, 10044 Stockholm, Sweden; melilli@kth.se (G.M.); minna@kth.se (M.H.); 2Department of Mechanical and Aerospace Engineering, Politecnico di Torino, C.so Duca dgeli Abruzzi 24, 10129 Torino, Italy; irene.carmagnola@polito.it (I.C.); chiara.tondaturo@polito.it (C.T.-T.); gianluca.ciardelli@polito.it (G.C.); 3POLITO BIOMed LAB, Politecnico di Torino, 10129 Turin, Italy; fabrizio.pirri@polito.it (F.P.); annalisa.chiappone@polito.it (A.C.); 4Department of Applied Science and Technology, Politecnico di Torino, C.so Duca degli Abruzzi 24, 10129 Torino, Italy

**Keywords:** hydrogel, methacrylated CMC, 3D printing, DLP

## Abstract

The development of new bio-based inks is a stringent request for the expansion of additive manufacturing towards the development of 3D-printed biocompatible hydrogels. Herein, methacrylated carboxymethyl cellulose (M-CMC) is investigated as a bio-based photocurable ink for digital light processing (DLP) 3D printing. CMC is chemically modified using methacrylic anhydride. Successful methacrylation is confirmed by ^1^H NMR and FTIR spectroscopy. Aqueous formulations based on M-CMC/lithium phenyl-2,4,6-trimethylbenzoylphosphinate (LAP) photoinitiator and M-CMC/Dulbecco’s Modified Eagle Medium (DMEM)/LAP show high photoreactivity upon UV irradiation as confirmed by photorheology and FTIR. The same formulations can be easily 3D-printed through a DLP apparatus to produce 3D shaped hydrogels with excellent swelling ability and mechanical properties. Envisaging the application of the hydrogels in the biomedical field, cytotoxicity is also evaluated. The light-induced printing of cellulose-based hydrogels represents a significant step forward in the production of new DLP inks suitable for biomedical applications.

## 1. Introduction

Additive manufacturing (AM) has rapidly changed from being a scientific interest to an industrially viable manufacturing process. The incredible success of 3D printing is mainly due to the possibility of building objects from 3D model data, with complex geometry, accurate resolution and relatively short execution time [1,2]. These intriguing advantages make 3D printing suitable for a variety of applications [3], ranging from the aerospace and automotive industries [4,5] to the biomedical field [6] in which the possibility of producing custom-made objects, models or devices can bring about enormous improvements [7,8,9,10]. Among the different 3D-printing technologies [11], light-assisted printing [12,13,14], including stereolithography (SL) [8] and digital light processing (DLP) [15,16], utilizes photopolymerization promoted by light irradiation to model complex structures [14]. DLP in particular shows good compromise in terms of high resolution (~1 m) and fast printing speed (mm^3^·S^−1^) for the production of unique 3D objects [17]. A common DLP/SL 3D-printable formulation is composed of a mixture of monomers (commonly acrylates), a light-sensitive initiator, and eventually, a dye. With the correct choice of ingredients and fillers for the photocurable formulation, structural [18,19,20,21,22] and functional [23,24,25,26,27] properties of the final objects can be tailored, opening the path for advanced applications. DLP and SL have been considered for bioprinting applications since the 2000s [28], and more recently have been proposed for the shaping of cell-laden photocurable inks [29]. There are extensive efforts toward the development of AM techniques and materials, but the supply of bio-based printing materials is still very limited. Bio-based photocurable formulations for light-assisted printing technologies are even rarer, but methacrylated silk fibroin [30], silk fibroin-polyethylene glycol [31], a mixture of gelatin methacrylate and glycidil methacrylate-hyaluronic acid [32], keratin [33], and a mixture of polyethylene glycol diacrylate and gelatin methacrylate hydrogel [34] have been reported. In a few cases, cellulose has been used in DLP processes, but it was added as a filler in the form of cellulose nanocrystals (CNCs) in synthetic acrylic monomer-based inks [35,36,37]. Lignocellulosic biopolymers and their derivatives present an abundant, largely unexplored source of materials with huge potential for light-assisted printing. Carboxymethyl cellulose (CMC) is a water-soluble cellulose ether which is commercially produced in large scale and used in a wide range of fields from food applications to paints and detergents [38,39]. In particular, because CMC is bio-compatible and FDA-approved, it is an attractive natural polymer for tissue engineering and regenerative medicine [40]. Moreover, CMC-based formulations can mimic cell microenvironments due to the similarity of CMC structure to the glycosaminoglycane component of extracelluar matrix [41]. CMC has already been proposed as a candidate 3D-printable material primarily using extrusion-based techniques [42,43]. Due to its versatility, its advantageous properties, water-solubility, and susceptibility to further functionalization, we also expected CMC would be an ideal candidate for the preparation of novel photocurable resins for DLP. However, the use of light-assisted printing techniques requires reactive photocrosslinkable functional groups, which means CMC needs functionalization to produce ink formulation for the production of 3D photocured hydrogels. CMC was therefore methacrylated and its photorheology and DLP printability was investigated in two formulations, namely, M-CMC/Dulbecco’s Modified Eagle Medium (DMEM) and M-CMC/water, in presence of a fixed amount of lithium phenyl-2,4,6-trimethylbenzoylphosphinate (LAP) photoinitiator. Envisaging the application of the hydrogels in the biomedical field, the mechanical properties, swelling behavior and preliminary in vitro tests with fibroblast cell line (NIH/3T3) were also evaluated.

## 2. Materials and Methods

### 2.1. Materials

Medium viscosity CMC sodium salt, methacrylic anhydride, and lithium phenyl-2,4,6-trimethylbenzoylphosphinate (LAP) were purchased from Sigma Aldrich. For a cytotoxicity test, DMEM purchased from Carlo Erba (Italy) was used.

### 2.2. Cellulose Functionalization

For the CMC modification, we followed a method similar to previously reported processes [44,45]. We dissolved 2.00 g of CMC sodium salt in 100 mL of water at 50 °C. The pH was periodically adjusted to 8.0 with 3 N sodium hydroxide (NaOH) for the entire duration of the reaction. After cooling the solution to 0 °C, 4 mL of methacrylic anhydride (MA) was added dropwise to the CMC solution and the reaction was continued for 24 h at 0 °C. The resulting mixture was precipitated and washed with ethanol to remove the unreacted methacrylic acid and methacrylic anhydride. After three days of dialysis against the water, the methacrylated product (M-CMC) was freeze dried for two days.

### 2.3. Hydrogel Preparation and Printing

A water solution of M-CMC (20 mg/mL) was prepared under magnetic stirring at 50 °C for 1 h. Afterwards, the solution was allowed to cool down, and 2 wt% (with respect to the M-CMC) of bis(acyl)phosphane oxi LAP was added.

The solution was placed in the vat of an Asiga UV-MAX DLP printer with an XY pixel resolution of 62 μm using a light-emitting diode light source (385 nm). Different CAD files were converted to STL files and 3D printed; the layer thickness was fixed at 50 μm and the light intensity to 30 mW/cm^2^, while the exposure time was varied for different prints. A post-curing process performed with a medium-pressure mercury lamp (6 min with a UV lamp provided by Robot Factory; light intensity 12 mW/cm^2^) followed the printing process.

### 2.4. Characterization

Attenuated total reflectance-Fourier transform infrared spectroscopy (ATR-FTIR) was used, first to confirm the successful methacrylation of CMC and then to evaluate the photopolymerization reaction. The experiments were conducted on dried samples by means of a Thermo Scientific Nicolet iS50 FTIR Spectrometer equipped with a diamond crystal ATR accessory. ATR spectra were collected with a resolution of 4 cm^−1^ in the range of 4000–600 cm^−1^.

All ^1^H NMR spectra were recorded by a Brucker Avance DPX-400 nuclear magnetic resonance spectrometer at 25 °C in D_2_O.

Real-time photorheological measurements were carried out to investigate the kinetics of photopolymerization of the M-CMC hydrogel; an Anton PAAR Modular Compact Rheometer (Physica MCR 302) in parallel-plate mode using a quartz bottom plate was used. A UV light source (Hamamatsu LC8 lamp, light intensity 25 mW·cm^−2^) was placed under the bottom plate. During the measurements, the gap between the two glass plates was set to 0.3 mm, and the sample was kept under a constant shear frequency of 1 Hz. Light was switched on after 30 s to allow the system to stabilize before the onset of polymerization. According to preliminary amplitude sweep measurements, all of the tests were carried out in the linear viscoelastic region (strain amplitude of 1%). Additionally, frequency sweep measurements (FS) were performed at a constant temperature (25 °C) from 0.01 to 10 Hz, setting the strain amplitude at 1%.

An SEM image was acquired by ultra-high-resolution FESEM (Hitachi S4800). The dried photocured M-CMC hydrogel was coated with a Pt/Pd coating at a thickness of 5 nm.

The fraction of material that had covalently crosslinked in the 3D-printed hydrogel network (gel content) was evaluated by first drying the sample in a vacuum oven (50 °C, 600 mbar), and then immersing it in distilled water at 40 °C for 24 h to leach the unreacted soluble fraction and/or the residues of initiator entrapped in the network. The sample was then dried in a vacuum oven (50 °C, 600 mbar) until constant weight. The amount of insoluble fraction was determined as the weight difference before and after extraction (%).

The water uptake was then evaluated in the hydrated printed samples. The samples were dipped in distilled water at 37 °C and the swelling kinetics were determined after taking out the soaked samples from the water at different time intervals and weighing them. The samples were then placed back into water, and the protocol was repeated until no further weight change was observed. The swelling degree (SD) was determined using the following equation:SD(%) = (Wt − W0)/W0 × 100
where W0 is the initial weight and Wt is the weight at established time. The results of the swelling degree and gel content were averaged over measurements carried out on three different samples.

The mechanical properties of 3D-printed hydrogels were evaluated through uniaxial compressive test performed using MTS Qtest/10 instrument equipped with 10 N cell load. Cylindrical hydrogels (h = 7 mm, φ = 6 mm) were printed and tested with no further treatments, setting a compression rate of 0.5 mm/min^−1^ until failure. The compressive modulus was determined as the slope of the curve in the linear range (strain < 20%). The value was averaged on 15 samples.

NIH/3T3 cells (ATCC^®^ CRL-1658™) were cultured in a 96-well plate using a cell density of 2 × 10^4^ cells/well. After 24 h, cells reached confluence and the culture medium was removed from each well and substituted with a conditioned medium prepared by soaking 0.1 g of printed hydrogel into 1 mL of DMEM. Control samples (CTRL) were prepared by substituting the medium with unconditioned fresh medium. After 24 h, the medium was substituted in each well with 100 μL of 0.1 mg·mL^−1^ non-fluorescent resazurin solution in phosphate-buffered saline (PBS). Cell viability measured as non-fluorescent resazurin was converted to fluorescent resorufin by cell metabolism and the fluorescent signal was monitored using a plate reader (Biotek) at 530 nm excitation wavelength and 590 nm emission wavelength. Experiments were performed using six samples for each condition and cell viability was calculated as a percentage value compared to CTRL.

## 3. Results and Discussion

Light-assisted printing, such as digital light processing, requires photocurable functional-groups. For this purpose, carboxymethyl cellulose was methacrylated. The methacrylation process was carried out at slightly basic pH, with methacrylic anhydride (MA) as methacrylation agent. In these conditions, the reaction of fully deprotonated alcohol groups of CMC and MA led to graft methacrylate functional groups distributed along the CMC backbone (M-CMC) (Figure 1A) [46]. The precise procedure is reported in the experimental section. Successful methacrylation of CMC was confirmed by FTIR and ^1^H NMR spectroscopy. FTIR and ^1^H NMR spectra of neat CMC were also collected for comparison.

The asymmetric (1590 cm^−1^) and symmetric (1413 and 1323 cm^−1^) stretching vibrations associated to the carboxylate (–COO–) were detected in both CMC and M-CMC FTIR spectra, respectively. A broad peak between 3000 and 3700 cm^−1^ due to the –OH stretching vibration was also detected, as well as the stretching vibration associated to –CH_2_–O–CH_2_– at 1060 cm^−1^ [45,47]. The methacrylated product, M-CMC, also showed the characteristic FTIR vibration bands derived from the –(C=O)O ester group, and the –C–H detected at 1715 and 811 cm^–1^, respectively (Figure 1B). The CMC methacrylation was further evaluated by ^1^H NMR. The ^1^H NMR spectrum of neat CMC was collected first for reference. The resonance of the CMC backbone was located between δ 2.7 and 5 ppm. The spectrum of M-CMC also showed peaks associated to vinyl (δ = 6.1 and 5.7 ppm) and methyl (δ = 1.9 ppm) groups, which confirmed the presence of methacrylate groups (Figure 1C).

The degree of substitution (DS) was calculated by the ratio between the total integral of the peaks in M-CMC related to –CH_3_ and –CH_2_ (associated with 5 hydrogen) and the total integral of the anhydrous glucose unit of CMC skeleton (associated with 8.4 hydrogen, in accordance with the substitute carboxymethyl groups). The calculated DS corresponds to 0.6, which indicates that more than 25% of the hydroxyl groups present in the CMC were effectively grafted with methacrylic groups.

After the methacrylation step, the photopolymerization reactivity of the functionalized material (M-CMC) was evaluated by photorheology. The freeze-dried M-CMC was solubilized in distilled water according to its maximum solubility (20 mg/mL). Envisaging the application of the studied hydrogel in the biomedical field, M-CMC was also solubilized in a culture medium (DMEM). In both formulations, a fixed amount of LAP photoinitiator was added to initiate the radical photopolymerization under UV light. In neutral conditions, DMEM presents light red coloration due to the phenol red molecules used as a pH indicator in the medium. Since the absorbance spectra of phenol red molecules and LAP are partially overlapped, the photoreactivity of the photoinitiator may be reduced, the rheological properties (storage modulus, G′, and loss modulus, G″) were monitored all along the UV-curing process for both formulations CMC/DMEM/LAP and M-CMC/water/LAP (Figure 2). The kinetics of the photocrosslinking reaction were recorded by turning on the light with 30 s delay. For both formulations, the reaction was almost complete within 60 s, which corresponds with the onset of the G′ plateau (Figure 2A). The intersection between the G′ and G″, defined as the gel point, is considered as the transition from liquid resin to solid polymer network. As shown in Figure 2A,B, both formulations achieved the gel point in less than 2 s, which confirms the high photoreactivity of M-CMC independent of the medium. Although the formulation M-CMC/DMEM/LAP showed a slight delay with respect to the onset of the curing process, the DMEM medium still allowed sufficient light penetration for the photocuring process in view of 3D printing.

After 90 s of UV irradiation, both formulations reached a stable G′ value, which corresponds with maximum methacrylic double-bond conversion. The storage modulus of hydrogels at the plateau (strain 1% frequency 1 Hz) was measured; hydrogels derived from M-CMC/water/LAP formulation showed an elastic modulus of 3320 Pa, while M-CMC/DMEM/LAP formulation provided hydrogels with a modulus of 1900 Pa. The moduli of the hydrogels derived from both formulations are extremely promising and in line with other DLP-3D-printable bio-based materials proposed in the literature at similar concentrations, such as methacrylated gelatin (Gel-MA) (20 mg/mL G′ ~3000 Pa) [48] and other chemically crosslinked CMC (G′ = 1030 Pa) [49] hydrogels mainly aimed to biomedical uses. The lower light penetration in the cell culture medium reduced the final conversion of the methacrylic groups, which in turn slightly reduced the storage modulus of the 3D-printed hydrogel. Furthermore, cations such as Na^+^, Ca^2+^ and Fe^3+^ (typically contained in DMEM medium) can interact with the carboxylate anions in CMC [50], reducing the mobility of the M-CMC chains and thus hindering the crosslinking reaction.

Frequency sweep measurements were also performed. Figure 2C reports the measurements done on the crosslinked hydrogel derived from M-CMC/DMEM/LAP formulation. As shown, G′ (elastic component of the modulus) is larger than G″ (viscous component) by a factor 15–20, which means a low phase angle (δ). The phase angle defines the viscoelastic behavior of the materials. Phase angles close to 0° correspond with elastic materials (solid behavior), while values close to 90° are related to viscous materials (liquid behavior). Thus, M-CMC hydrogel behaves like a solid with elastic properties (δ = 1.5–4°). This test also indicated good stability for the material, which was not broken at frequencies of up to 10 Hz.

The photorheology and rheology results provided useful information (e.g., storage modulus, photoreactivity of the formulations, gel point, etc.) for testing the material with a DLP 3D printer. Thus, different digital models were designed and converted to STL files.

Through the DLP 3D printer, simple structures like cylinders, parallelepipeds (Figure 2A) as well as more complex geometries (Figure 3D–F) were successfully fabricated from M-CMC/DMEM/LAP and M-CMC/water/LAP formulations, respectively. The 3D-printed hydrogels appeared mechanically stable and flexible (Figure 3B). The obtained hydrogels were further subjected to a post-curing process. The treatment time was adjusted based on the previous photorheology results in order to achieve a completed photocrosslink reaction.

Photocurable printable formulations usually require the use of dyes that absorb in the range of the light source emission. Although the presence of the dyes reduces the photoinitiator light absorption, they have the important role of limiting light diffusion in the liquid formulation, ensuring a good resolution of the printed part. In M-CMC/water/LAP formulation, no dye was used during the printing process. Thus, the 3D shaped hydrogel appeared transparent and was characterized by low resolution. Such formulation (without the addition of the dyes) is suitable for the production of massive parts with low resolution that do not require voids along the *z*-axis. Phenol red molecules contained in the M-CMC/DMEM/LAP formulation may act as a dye improving the printability and resolution of more complex 3D-printed structures. As visible in Figure 3E,F, suspended structures could be built, as well as thin walls with sub-millimetric details. In the present work we chose to avoid the use of different dyes commonly used during 3D printing because of the potential cytotoxicity, which still needs to be assessed.

The SEM image (Figure 3C) of a freeze-dried sample shows a classical morphology for hydrogels presenting micro-porosity. Such micro-structures will also influence the macroscopic properties of the M-CMC hydrogel.

The 3D-printed hydrogels prepared from M-CMC/water/LAP formulation were then subjected to further characterizations. The determination of the crosslink degree of the hydrogel was evaluated by extracting the soluble unreacted M-CMC from the crosslinked hydrogel and dipping the samples in water for 24 h at 40 °C. The insoluble crosslinked M-CMC fraction was calculated to be 87 wt%. Considering the low concentration of the M-CMC in water (20 mg/mL), the extracted crosslinked fraction reveals a good reactivity of the M-CMC chains during the printing process. FTIR spectra were collected to further confirm the effective crosslinking reaction. The FTIR spectrum of the freeze-dried M-CMC powder was collected to correlate the changes in the vibration modes of the photocured printed samples. The consumption of methacrylic double bonds, involved in the crosslinking reaction, was indicated by the disappearance of the –C–H band at 811 cm**^−^**^1^ (Figure 4A).

An M-CMC/water/LAP formulation was used to print 3D cylindrical shaped hydrogels that were used for the evaluation of the compressive elastic modulus of the hydrogels.

Figure 4B reports one representative stress–strain curve; the value of the compressive modulus was calculated in two ranges of deformation, according to the method reported by Buyanov et al. [51]. The slope of the first linear region correspond to a Young’s modulus of 32 kPa, while increasing the imposed deformation, the modulus increases up to 84 kPa. These values are in line with other bio-based chemically crosslinked hydrogels presented in the literature composed of gellan gum which allow the encapsulation of NIH-3T3 fibroblasts [52] or GelMA [53]. The obtained modulus calculated at the strain of 25% is also on the same order of magnitude observed for DLP-printed silk fibroin hydrogels (20 mg/mL E ~20 kPa, 30 mg/mL E ~50 kPa) on which cell viability was tested envisaging the application in cartilage tissue engineering [30].

The swelling ability of the DLP 3D-printed M-CMC hydrogel was also evaluated. The possibility to further incorporate water in the hydrated state after printing, without breaking or deforming, was also assessed, showing the great potential of this material. The printed hydrogels were able to absorb up to 650% of their initial weight in water (Figure 4C). Figure 4D shows the increase of the dimensions of a 3D-printed parallelepiped print after 24 h in water. The swelling kinetics show a rapid diffusion of water in the M-CMC hydrogel (Figure 4C). Within 150 min the uptake reaches 90% of the total amount of absorbed water. The swelling ratio is a combined effect of morphology (Figure 3C) and chemical structure of the hydrogel. The strong electrostatic repulsion of the remaining anionic units (-COO-) along the CMC chains [54] coupled with the microporous structure explains the high swelling ratio at equilibrium (SD = 651; t = 24 h) achieved by the M-CMC hydrogel. Such a water uptake by the printed sample corresponds to a weight increase from the theoretical dry content of more than 35,000%; similar values were previously observed for superabsorbent hydrogels based on crosslinked CMC and starch [55].

Finally, the 3D-printed hydrogels solubilized in the culture medium underwent preliminary cytotoxicity testing to assess the lack of cell death induced by the release of LAP photoinitiator or unreacted polymer chains within the first 24 h [56]. No cytotoxic events were measured and the cell viability of NIH/3T3 was comparable to control conditions, obtaining a viability of 96.7 ± 5.2% compared to control (100%).

## 4. Conclusions

The development of bio-based photocurable materials suitable for 3D printing is a stringent request for the expansion of this brilliant production technique. Here, a new ink for DLP printing starting from lignocellulosic-based materials was successfully pursued. A methacrylated-CMC water-based formulation was 3D-printed, and the resulting hydrogel presented extremely promising mechanical and swelling properties. Furthermore, the preliminary cytotoxicity tests confirmed its potential in the biomedical application field.

## Figures and Tables

**Figure 1 polymers-12-01655-f001:**
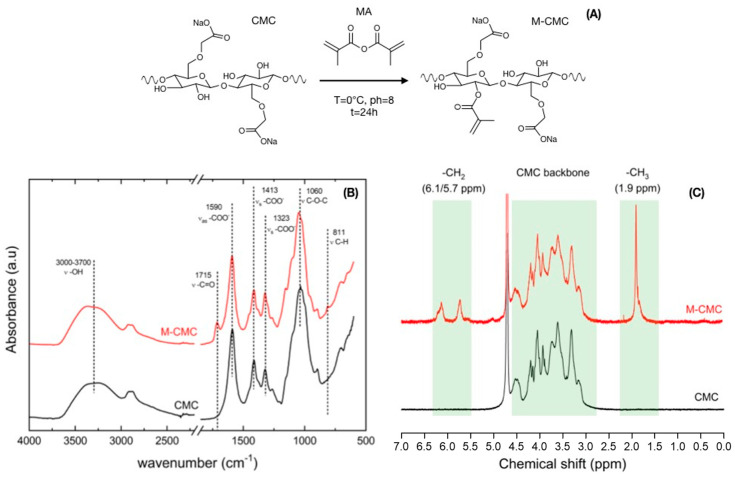
(**A**) Scheme for the methacrylation of carboxymethyl cellulose (CMC). The presented product only presents one of possible reaction products. (**B**) FTIR and (**C**) ^1^H NMR spectra for methacrylated CMC (M-CMC, red) and neat CMC (CMC, black).

**Figure 2 polymers-12-01655-f002:**
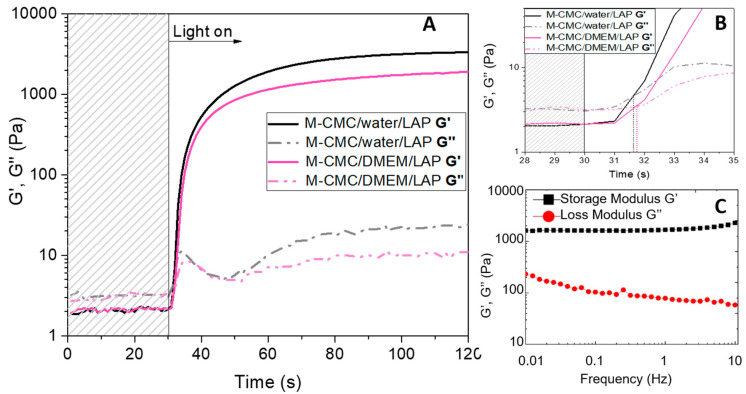
(**A**) Photorheology of methacrylated carboxymethyl cellulose (M-CMC) 20 mg/mL (2 wt% lithium phenyl-2,4,6-trimethylbenzoylphosphinate (LAP)) solubilized in water (black) or in culture medium (pink). (**B**) Gel point. Film thickness 300 m. (**C**) Frequency sweep. Strain rate 1% and oscillation frequency from 0.01 to 10 Hz.

**Figure 3 polymers-12-01655-f003:**
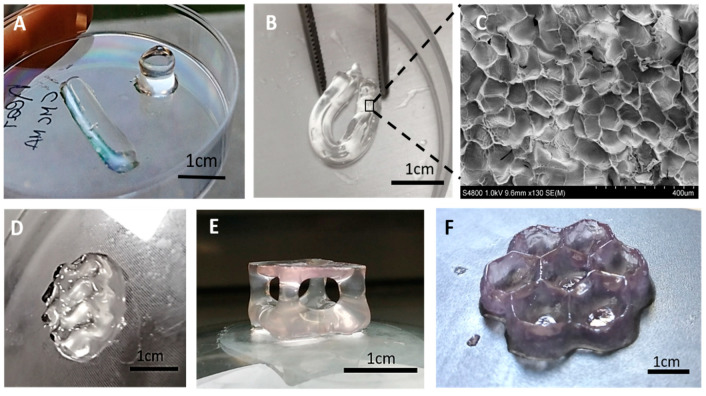
3D-printed M-CMC hydrogels. (**A**) Simple cylinders and parallelepipeds (solvent: water). (**B**) The hydrogel exhibited good flexibility and handleability. (**C**) SEM analysis performed on the freeze-dried hydrogel. (**D**–**F**) 3D objects printed from water (**D**) and from culture medium solution (**E**,**F**).

**Figure 4 polymers-12-01655-f004:**
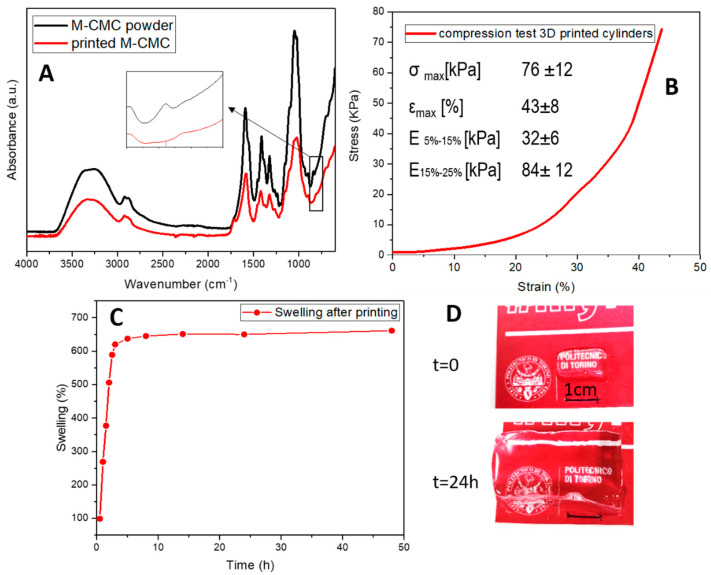
(**A**) FTIR spectra of the 3D-printed M-CMC hydrogel, (**B**) Stress–strain curve obtained from a compression test performed on 3D-printed cylinders. (**C**,**D**) Swelling of the 3D-printed hydrogel as printed.

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
