# Peer review of "DLP 3D Printing Meets Lignocellulosic Biopolymers: Carboxymethyl Cellulose Inks for 3D Biocompatible Hydrogels"

_polymers, 2020, doi:10.3390/polym12081655_

Round 1
Reviewer 1 Report
The paper investigates the use of methacrylated carboxymethyl cellulose (M-CMC) for Digital Light Processing (DLP) 3D printing. The results are interesting for the biomedical field as new materials for DLP is a hot topic. I understand that the authors do not try to use the ink as a bioink, and just as a material for 3D printing.
My main concern is located in the cytotoxicity studies. I would like to see some references to previous works from other authors using conditioned medium prepared by soaking 0.1 g of printed hydrogel into 1 mL of medium. I am not sure that this is the best procedure. The authors should present a longer incubation time of the cell in contact with the hydrogel to have more realistic results. In my own XP, the viability of 3T3 fibroblasts of 96.7 ± 5.2% is too optimistic, which could mean that the protocol is not the right one to compare with other works. Please discuss this issue using previous publications not only on CMC but also other cellulose-based materials or other biomaterials.
Author Response
The response are reported in the attached file.

Reviewer 2 Report
The authors report on the development of a stereolithography compatible hydrogel based on carboxymethyl cellulose (CMC). Methacrylation of CMC with an added photosensitizer/initiator allowed 3D fabrication use a DLP light engine printer. The chemistry is straightforward, the characterization of the process and material is fairly comprehensive, and the printed materials show properties that are of interest to the bioprinting community. A well written paper, I recommend publication following some minor revision.
Cell viability is only mentioned in the last sentence. This may be fine, but I recommend showing the data (supporting info perhaps) as the comparison is made against a control that is not entirely clear.
The authors should note in the introduction or results that CMC is a candidate 3D print material primarily using extrusion. A couple examples:
https://doi.org/10.3390/ma11030454
https://doi.org/10.1016/j.matdes.2018.09.009
Also useful to note in the intro is that use of DLP (DMD devices for bioprinting) has been considered for some time. Examples:
https://doi.org/10.1002/jbm.a.30601
https://doi.org/10.1002/smll.200801084
Author Response
The answer to the reviewer are reported in the attached file

Reviewer 3 Report
In the manuscript, the authors attempted to synthesize a photocurable M-CMC hydrogel to serve as a candidate ink for DLP-based bioprinting. While overall well-written and presented with appropriate graphical presentations, the manuscript has currently several limitations that need to be thoroughly addressed before it could be recommended for publication.
- According to the stress-strain curve shown in Fig. 4B, the compressive modulus should be remarkably lower than the value the authors claimed in the manuscript since about 3-4 kPa was measured at the strain of 20%.
- In line 262, the content of the literature (ref. 47) the authors described is wrong. According to the literature, Shinn et al. developed a two-step process to fabricate the GelMA-reinforced gellan gum-MA (GGMA) hydrogel by infiltrating the GelMA solution (5%-20% or 50-200 mg/mL) into a pre-photocured GGMA hydrogel. Regarding this, the second network the author indicated should be GelMA. A Similar problem is also observed in line 198 of which the G’ of 2% silk fibroin was measured without UV-curing.
- In line 117, the authors indicated that the crosslinking assessment was performed by soaking the sample in DI water at 40°C for 24 h. However, in line 248, the soaking temperature the authors used was 37° Please confirm it.
- The section of “Discussion” is missing in the manuscript. If the authors attempted to combine this section with “Results”, the subtitle should be changed accordingly.
- The proposed hydrogel possesses extremely high water adsorption capacity (350000%), implying that the shape of the as-printed construct should remarkably swell after implantation. Under this circumstance, the authors should clarify its applicability in tissue engineering or biomedical applications.
- Although the cytotoxicity of the hydrogel was assessed according to ISO-10993-5. The interaction between the hydrogel and cells should be evaluated if the target application of the hydrogel is to serve as bioinks. Thus, it is highly recommended to involve the in vitro experiments that could be used to evaluate the growth of cells cultured on or in the cured hydrogel in the manuscript.
- Some minor issues about the subscript and superscript letters should be thoroughly checked and revised in the manuscript. Additionally, the citation numbers should in the reference list needs to be corrected.
Author Response
The answer to the reviwer are reported in the attached file

Round 2
Reviewer 1 Report
No additional comments.
Author Response
Thank you for the reviewer comments.
Reviewer 3 Report
The authors have answered the critical issues I drew. Thus, it is recommended to consider this article for publication in the "Polymers". However, some minor issues should be corrected before accepted.
- In line 41, "mm3s-1" should be corrected to "mm3s-1"
- In line 45, the abbreviation should be changed to "SL" instead of "SLA" since the authors had defined it in line 38.
- The concentration unit (mg·mL-1 or mg/mL) should be unified.
- In line 272, .....resulted to corresponded to....
Author Response
We have corrected the minor revision requested by the reviewer in the second revision of the paper.